# *Nna1*, Essential for Purkinje Cell Survival, Is also Associated with Emotion and Memory

**DOI:** 10.3390/ijms232112961

**Published:** 2022-10-26

**Authors:** Li Zhou, Kohtarou Konno, Maya Yamazaki, Manabu Abe, Rie Natsume, Masahiko Watanabe, Hirohide Takebayashi, Kenji Sakimura

**Affiliations:** 1Center for Coordination of Research Facilities, Institute for Research Promotion, Niigata University, Niigata 951-8585, Japan; 2Department of Animal Model Development, Brain Research Institute, Niigata University, Niigata 951-8585, Japan; 3Department of Anatomy, Faculty of Medicine, Hokkaido University, Sapporo 060-8638, Japan; 4Division of Neurobiology and Anatomy, Graduate School of Medical and Dental Sciences, Niigata University, Niigata 951-8510, Japan

**Keywords:** *Nna1/CCP1/Agtpbp1*, nuclear localization signals, AMPA, kinesin-1, less anxiety

## Abstract

*Nna1/CCP1* is generally known as a causative gene for a spontaneous autosomal recessive mouse mutation, Purkinje cell degeneration (*pcd*). There is enough evidence that the cytosolic function of the zinc carboxypeptidase (CP) domain at the C-terminus of the Nna1 protein is associated with cell death. On the other hand, this molecule’s two nuclear localization signals (NLSs) suggest some other functions exist. We generated exon 3-deficient mice (*Nna1^N^* KO), which encode a portion of the N-terminal NLS. Despite the frameshift occurring in these mice, there was an expression of the Nna1 protein lacking the N-terminal side. Surprisingly, the *pcd* phenotype did not occur in the *Nna1^N^* KO mouse. Behavioral analysis revealed that they were less anxious when assessed by the elevated plus maze and the light/dark box tests compared to the control. Furthermore, they showed impairments in context-dependent and sound stimulus-dependent learning. Biochemical analysis of *Nna1^N^* KO mice revealed a reduced level of the AMPA-type glutamine receptor GluA2 in the hippocampal synaptosomal fraction. In addition, the motor protein kinesin-1, which transports GluA2 to dendrites, was also decreased. These results indicate that *Nna1* is also involved in emotion and memory learning, presumably through the trafficking and expression of synaptic signaling molecules, besides a known role in cell survival.

## 1. Introduction

*Nna1* (also known as *CCP1* or *Agtpbp1*) was identified as a causative gene for Purkinje cell degeneration (*pcd*), an autosomal recessive disorder [1]. Defects in *Nna1* also cause degeneration not only in Purkinje cells but also in many other neurons of the central nervous system, including cerebellar granule cells, neurons in the deep cerebellar nuclei, neurons in the inferior olive, as well as retinal photoreceptors, olfactory bulb neurons, and individual subpopulations of thalamic neurons [2,3,4,5,6]. Another report shows that mice deficient in this molecule exhibit male infertility due to defective spermatogenesis [7].

The mouse *Nna1* gene encodes a protein of 1218 amino acids (aa), and the molecule contains multiple functional domains. The most investigated region is a zinc carboxypeptidase (CP) domain located in the C-terminus (843–1013aa); mutations in this region induce *pcd* [8,9]. There are two nuclear localization signals (NLSs) present at the N and C termini (144–151aa and 996–1016aa) [8,10], and studies using fusion proteins with green fluorescent protein have shown that Nna1 exists in both the nucleus and cytoplasm [10]. We have so far conducted various examinations to elucidate the function of the Nna1 protein in the nervous system [11,12,13]. To pursue the function of the CP domain, we generated and analyzed mutant mice lacking exons 21 and 22 that encoded the domain and reproduced the *pcd* phenotype. However, Western blot analysis using two anti-Nna1 antibodies against two C-terminal sites (837–1186 aa and 1188–1218 aa) showed that multiple Nna1 bands present in WT brains were completely undetectable in mutant brains [14]. Analysis of *Nna1* mRNA by Northern blot and in-situ hybridization revealed the absence of the transcripts, likely due to degradation by nonsense-mediated mRNA decay (NMD, [15]), thus being regarded as *Nna1^null^* or KO mice.

In this study, we attempted to generate mice lacking exon 3, which encodes a part of the NLS domain at the N-terminal side, to elucidate the function of the N-terminal side of this molecule. In the *Nna1* N-terminal knockout mice (*Nna1^N^* KO), we observed the expression of N-terminus-truncated Nna1 protein and no apparent ataxia, in contrast to the *pcd* mice. Behavioral analyses of *Nna1^N^* KO mice showed impaired context- and sound cue learning, and less anxiety than WT mice. Furthermore, biochemical analysis revealed that AMPA-type glutamate receptor GluA2 was reduced in the synaptosomal fraction. These results suggest that the *Nna1* is involved in higher brain functions, such as emotional behavior and memory.

## 2. Results

### 2.1. Generation of Nna1^N^ (Nna1^ΔEx3^) KO Mice

Assuming that *Nna1* mRNA has multiple transcription or translation start sites, we established mice floxed with exon 3 (Figure 1a,b), which encodes a part of the NLS located on the N-terminal side of *Nna1*. These mice were crossed with *TLCN-Cre* mice to generate exon 3-deficient mice (Figure 1a,c,d). We named these exon 3-deficient mice and previously generated exons 21, 22 deficient mice, *Nna1^N^* KO and *Nna1^C^* KO mice, respectively. To examine Nna1 expression in the cerebral cortex, cerebellum, and hippocampus of these mice, we performed Western blot analysis using the anti-Nna1 antibody against the C-terminus (1188–1218 aa). Several ladder-like bands, including a 150 kD band, were detected in WT mouse brains, while in *Nna1^C^* KO mice, these bands were barely detected (Figure 1e). In the *Nna1^N^* KO mice, a weak Nna1 band slightly smaller than the 150 kDa and ladder-like bands were observed. Northern blotting using the *Nna1* C-terminal probe (exons 17 to 23) revealed stronger signals in the cerebrum, cerebellum, and hippocampus of *Nna1^N^* KO mice than in WT mice (Appendix A). Since deletion of the exon 3 of the *Nna1* gene causes a frameshift, the presence of truncated Nna1 proteins in the *Nna1^N^* KO mice suggests that *Nna1* mRNA has multiple translation start sites and escaped from NMD. The increased amount of mRNA in *Nna1^N^* KO mice could be a compensatory upregulation for the loss of intact Nna1 proteins.

### 2.2. Normal Morphology in Nna1^N^ KO Mice

We next performed morphological analysis to investigate the brain phenotype of *Nna1^N^* KO mice. Macroscopic images of the adult brains showed no cerebellar atrophy seen in *Nna1^C^* KO and pcd mice (Figure 2a). We next performed histological analyses on the cerebellum by Calbindin-D28K immunohistochemistry, and there were no differences in the lobular and laminar structures between WT and *Nna1^N^* KO mice (Figure 2b,c). Double staining for Car8 and VGluT1, markers in Purkinje cells and parallel fiber terminals, respectively, showed no discernible differences in dendritic arborization and the spine formation of Purkinje cells and in synapse formation with parallel fibers on distal spiny branchlets (Figure 2d,e). Immunohistochemistry for VGluT2, a marker for climbing fiber terminals, indicated no significant differences in the distribution and wiring of climbing fiber synapses on proximal shaft dendrites (Appendix A–c). In addition, Nissl staining of the hippocampus showed no significant differences in the histology and cellular alignment between WT and *Nna1^N^* KO mice (Figure 2f,g).

### 2.3. Nna1^N^ KO Mice Are Impaired in Emotional and Memory Learning

In the open-field test, *Nna1^N^* KO mice were more hyperactive and spent much more time in the center of the open-field area compared with WT mice (Figure 3a–c), while there was no significant difference in their movement speed (Figure 3d). In the next light-dark transition test, *Nna1^N^* KO mice made a more significant number of transitions between the light and dark areas than WT mice (Figure 4a–c). However, there was no significant difference in the total distance traveled (Figure 4b,d). There was no significant difference in the time spent in the light area of the box between WT and *Nna1^N^* KO mice, but *Nna1^N^* KO mice seemed to spend more time in the dark area with more motility than the wild-type mice, indicating a trend of hyperactivity (Figure 4b,e). Furthermore, we performed the elevated plus-maze test to evaluate anxiety (Figure 5a). Although there was no significant difference in the total distance traveled (Figure 5b), *Nna1^N^* KO mice spent more time in the open arm of the maze and the center of the crossed arms than WT mice, showing less anxiety in *Nna1^N^* KO mice (Figure 5c,d).

Finally, we performed the contextual- and cued-fear conditioning tests to examine the effect of N-terminal deletion of *Nna1^N^* KO on learning memory. Adult mice aged 8–10 weeks were used for the conditioning test on day 1, the contextual test on day 2, and the cued test on day 3 (Figure 6a). For the conditioning test, mice were allowed to explore freely for 3 min and then presented with 55 dB white noise as a conditioned stimulus (CS) for 20 s, including the last 2 s of a foot shock (0.2 mA, 2 s) as an unconditioned stimulus (US). Then similar stimulation, 60 s of CS, including the last 2 s of US, was repeated twice (Figure 6b). There was no noticeable difference in response to the stimuli between WT and *Nna1^N^* KO mice (Figure 6c). However, in a spatial-dependent learning test, there were considerable differences in freezing behavior between WT and KO mice (Figure 6d,e). Furthermore, in a sound-dependent learning test conducted the next day, the freezing rate in *Nna1^N^* KO mice was lower than in WT mice (Figure 6f,g).

### 2.4. Nna1^N^ KO Mice Have a Different Subunit Composition of Glutamate Receptors in the Hippocampus

Since hippocampal function is known to be associated with memory deficits, we examined AMPA-type glutamine receptor subunit expression, which is closely related to hippocampal-dependent behavior [16]. Immunohistochemical analysis revealed increased tendency of GluA1 and GluA2 expressions (Figure 7a). Western blots of crude hippocampal lysates indicated significantly increased GluA2. At the same time, GluA1 showed an increasing trend but no significant difference (Figure 7b). Interestingly, polyglutamylated tubulin, whose side chains are shortened by the peptidase activity of Nna1 protein, was significantly increased (Figure 7c, *p* < 0.05). Furthermore, we observed a mild increase in GluA2 in the *Nna1^N^* KO cerebellum (Appendix A). Since a change in AMPA-type glutamine receptors content in the synapses may lead to changes in synaptic transmission in the *Nna1^N^* KO hippocampus, we measured the GluA1 and GluA2 in the synaptosome fraction. Surprisingly, there was a significantly decreased concentration of GluA2 in the synaptosome fraction of the *Nna1^N^* KO hippocampus, whereas there was no change in GluA1 (Figure 7d). Furthermore, when kinesin-1 was measured, which is involved in the transport of GluA2-containing vesicles, its concentration was higher in the crude fraction and lower in the synaptosome fraction, like GluA2 (Figure 7d,e). These results suggest that in *Nna1^N^* KO mice, there are defects in the transport of GluA2 vesicles by kinesin-1 complexes.

## 3. Discussion

In this study, we generated *Nna1^ΔEx3^* KO (*Nna1^N^* KO) mice, in which the N-terminal exon 3 of *Nna1* was deleted to analyze the phenotype. We found that *Nna1^N^* KO mice showed no ataxia as seen in *Nna1* null [14] or *pcd* mice lines and no difference in body weight compared to WT mice. *Nna1* mRNA is frameshifted by the exon 3 deletion and was expected to be degraded by NMD. However, Northern blot analysis using the *Nna1* probe (exons 17–23) showed two prominent bands, similar to the WT (Appendix A). Western blotting with a Nna1 antibody against the C-terminal region showed multiple bands with a maximum molecular weight of approximately 150 kD in the WT mice. In the *Nna1^N^* KO, there was a band with slightly lower molecular weight than the WT band (~150 kD) and multiple bands in a ladder-like pattern (Figure 1). In the *Nna1^C^* KO mice lacking exon 21, 22 encoding carboxypeptidase domain, neither *Nna1* mRNA nor protein was detected [14], indicating that mRNA degradation occurs in the *Nna1^C^* KO mice by NMD. However, in the *Nna1^N^* KO mice, the amount of *Nna1* mRNA was increased in some brain regions and truncated Nna1 protein was observed. These findings raise some possibilities: (1) the *Nna1* gene has multiple sites for transcription start, (2) multiple isoforms with exon skipping, and (3) *Nna1* mRNA has multiple sites for translation start. Furthermore, the *Nna1^N^* KO mice lack the N-terminal side of the Nna1 protein but retain the carboxypeptidase domain of the C-terminal region, enabling us to evaluate the function of the N-terminal side of the Nna1 protein. Notably, we cannot exclude the possibility that the low expression of the truncated Nna1 proteins, in addition to the loss of the N-terminus of the Nna1 protein, contributes to the phenotype of the *Nna1^N^* KO mice.

Although the *Nna1^N^* KO mice showed no severe cerebellar phenotypes, such as Purkinje cell death, they exhibited a variety of phenotypes in systematic behavioral analyses, such as hyperactivity, reduced anxiety-like behavior, and impaired memory learning. These findings indicate that *Nna1* is involved not only in neuronal survival but also in higher brain functions related to emotion and memory. Furthermore, both contextual and cued fear conditioning tests showed impaired learning ability, suggesting that *Nna1* was involved in altered synaptic transmission in the hippocampus (Figure 7d). Thus, we investigated the expression level of AMPA-type glutamate receptors in the hippocampus. Immunohistochemistry showed that GluRA1 and GluRA2 were elevated, while GluRA3 and GluRA4 were unchanged (Figure 7a); detailed Western blotting analysis revealed that only GluA2 was significantly increased in the hippocampal crude fraction. In the synaptosomal fraction, GluA2 was significantly decreased in concentration, but GluA1 was unchanged. We next measured the amount of kinesin-1 that is involved in AMPA receptor transport [17] and found that, as with GluA2, its concentration was high in the crude fraction and low in the synaptosome fraction (Figure 7b–e), suggesting that there is some impairment occurring in kinesin-1 and GluA2 complex transport. Therefore, we examined the amount of polyglutamylated tubulin, which is associated with *Nna1* [18], and found that polyglutamylated tubulin was significantly increased (Figure 7b,c). These findings suggest that Nna1 is involved in GluA2 transport to the synapse by regulating the polyglutamine content of tubulins. Notably, a recent study has reported patients with global developmental delay and hypotonia with a novel homozygous c.3293G>A mutant of the *NNA1* gene in a consanguineous family [19].

The post-translational modifications (PTMs) of tubulin occur in the microtubules (MTs) of neurons and play essential roles in the dynamics and organization of MTs, thus exerting a direct effect on cell functioning, which is critical in human health and disease [20]. Nna1 is a cytoplasmic carboxypeptidase that modifies the C-terminal tail of tubulins, such as polyglutamylation, detyrosination, and generation of Δ2-tubulin [21,22]. The carboxy-terminus of tubulin is polyglutamylated and is exposed on the outer surface of tubulin during microtubule assembly. The length of the polyglutamate side chain on tubulin is vital for neuronal stability and survival [23]. In this study, we showed that the hippocampus of *Nna1^N^* KO mice expresses higher levels of polyglutamylated tubulin than WT mice (Figure 7b). This observation indicates that the N-terminal loss may result in reduced polyglutamate pruning of Nna1. In addition, the “modified” tubulin rail, where motor protein kinesin-1 binds, makes it difficult for kinesin-1 to move, presumably related to the reduced transport rate of GluA2.

There are two possible reasons for the increased expression of GluA2 in the hippocampal homogenates in the *Nna1^N^* KO mice. One is that the transport of GluA2 from the soma to the synaptic terminals is reduced due to a dysfunction in the axon transport and accumulates in the soma; N-terminal loss could ultimately reduce GluA2 transport to the synapse and induce qualitative changes in synaptic transmission. Another is that the decrease in GluA2 in synaptosomes may provide feedback to the soma to produce more GluA2 and compensate for the decrease in the synapse terminals. The present study demonstrates that *Nna1* functions not only in neuronal survival but also in emotion and memory, possibly via synaptic transmission by polyglutamate modification of tubulins.

## 4. Materials and Methods

### 4.1. Construction of the Nna1 Conditional Allele

A bacterial artificial chromosome (BAC) clone containing the *Nna1* gene (RP-23-119N9) was used to construct targeting vectors. The nucleotide sequence of the mouse genome was obtained from the National Center for Biotechnology Information (NCBI; Genbank accession number: NC_000079) to construct the *Nna1*-targeting vector. The 5′ homology arm of 5.83 kb (5523–11356; the nucleotide residues from the mouse BAC clone are numbered in the 5′ to 3′ direction, beginning with the A of ATG, the initiation site of translation in *Nna1*, which refers to position +1, and the preceding residues are indicated by negative numbers) and the 3′ homology arm of 6.83 kb (12165–19000) from the BAC clone were subcloned into the pD5UE-2 L4-R1 and pD3DE-2 R2-L3 vectors (Invitrogen, Carlsbad, CA, USA), respectively, using a counter-selection BAC modification kit (Gene Bridges, Dresden, Germany). The 807 bp DNA fragment (11357–12164) containing *Nna1* exon 3 was amplified by PCR and subcloned between two loxP sites of a modified pDME-1 L1-L2 vector containing a phosphoglycerate kinase (*pgk*) promoter-driven neomycin cassette (pgk-neo) flanked by two FLP recognition target (FRT) sites. These three plasmids were directionally subcloned into pDEST-DT R4-R3 containing the diphtheria toxin gene (MC1-DTA) using LR clonase from a MultiSite Gateway Three-Fragment Vector Construction kit (Invitrogen) to yield the targeting vector. The targeting vector linearized with *Not*I was electroporated into the embryonic stem (ES) cell line, RENKA [24], which was derived from the C57BL/6N strain (MGI:5657107, Charles River Japan). ES cells were cultured on mitomycin C-treated neomycin-resistant fibroblasts in KnockOut Dulbecco’s modified Eagle’s medium (Cat.#10829018, Thermo Fisher Scientific, Waltham, MA, USA) supplemented with MEM non-essential amino acids (Cat.#11140050, Thermo Fisher Scientific), 17.7% KnockOut Serum Replacement (Cat.#10828028, Thermo Fisher Scientific), 884 μM sodium pyruvate (Cat.#S8636, Sigma, St. Louis, MO, USA), 88.4 μM 2-mercaptoethanol (Cat.#M6250, Sigma) and 884 U⁄mL murine leukemia inhibitory factor, ESGRO (Cat.#ESG1107, Merck). We introduced approximately 30 μg of vector DNA into 2 × 10^6^ ES cells in a 4 mm electroporation cuvette (Cat.#165-2088, BioRad Laboratories, Hercules, CA, USA) using Gene Pulser Xcell™ Electroporation Systems (Cat.#165-2661, BioRad Laboratories). G418 selection (175 g/mL) was started 38–48 h after electroporation and continued for 1 week. We identified three correctly targeted clones from G418-resistant clones (total 192 clones) by Southern blot analyses using the 5′ outer probe (5159–5522) on *Sca*I, the 3′ outer probe (19001–19492) on *Spe*I, and the *neo* probe [25] on *Spe*I-digested genomic DNA. PCR primers to amplify DNA fragments for the generation of ^32^P-labeled 5′ outer probe and 3′ outer probe are listed in Table 1.

### 4.2. Generation of Nna1^N^ Knockout (KO) Mice

Male chimera mice were generated by injection of recombinant ES cells into eight-cell stage embryos from ICR mice (MGI:5462094, SLC Japan), and then heterozygous mice (*Nna1^flox(neo+)^*) were obtained by natural mating with C57BL/6N female mice (MGI:5657107, Charles River Japan). To generate a *Nna1* knockout allele lacking exon 3 (*Nna1^ΔEx3^*), heterozygous F1 mice were crossed with *TLCN-Cre* mice (MGI:3042494) [26,27], which ubiquitously express Cre recombinase. Double heterozygous mice (*TLCN-Cre*; *Nna1^flox(neo+)^*) were crossed with C57BL/6N mice to generate the *Nna1^N^* KO allele (*Nna1^ΔEx3^*). Finally, *Nna1^N^ KO* mice (*Nna1^ΔEx3/ΔEx3^* mice) were generated by crossing heterozygous pairs (*Nna1^ΔEx3/wt^* mice). All mice used in this study were maintained in the C57BL/6N background. Animal care and experimental protocols were approved by the Animal Experiment Committee of Niigata University and were carried out under the Guidelines for the Care and Use of Laboratory Animals of Niigata University (approved number: SA00733, SA01091). Animals were handled under the guidelines established by the Institutional Animal Care and Use Committee of Niigata University. The following measures were taken to minimize animals’ suffering during experiments: restlessness, vocalizing, loss of mobility, failure to groom, open sores/necrotic skin lesions, guarding (including licking and biting) a painful area, and a change in body color. If these signs were observed, they were excluded from further participation and treated appropriately according to the approved protocol. The mice used were 3–50 weeks old and had 8–29 g body weight. Time-point of each experiment was described in each result. No randomization was performed in this study.

### 4.3. Genotyping PCR

Genotyping by PCR was performed as follows. Genomic DNA was extracted from the tips of the tails of wild-type and *Nna1* mutant mice: the tail tissues were incubated with 0.025 N NaOH and 2 mM EDTA for 30 min at 100 °C and then mixed with an equal volume of 40 mM Tris-HCl (pH 8.0) at around 20 °C. The extracted DNA was used as a template for the PCR reaction using the *Nna1*-lox forward and *Nna1*-lox reverse primers (Table 1). PCR was performed using Quick Taq HS Dye Mix (Toyobo, Osaka, Japan) under the following PCR conditions: 95 °C for 30 s, 30 cycles of 95 °C for 10 s, 60 °C for 30 s, and 72 °C for 1 min, followed by 72 °C for 5 min. The PCR products were separated by agarose gel electrophoresis to identify the DNA bands; 1050 bp and 560 bp were amplified from the WT and *Nna1* KO allele, respectively.

### 4.4. Northern Blot Analysis

Northern blot analysis was performed as previously described [28]. Briefly, Poly(A) -RNAs (10 μg) were extracted from a different part of the mouse brain under the conditions described above, electroporated with agarose-formaldehyde gel, and transferred to a nitrocellulose membrane (Schleicher and Schuell). The membranes were washed in 3 × SSC and dried in air followed by UV link, then hybridized with 32P-labeled *Nna1* cDNA probes for 18 h at 42 °C in 50% formamide, 5 × SSC, 1 × Denhardt’s solution, 20 mM sodium phosphate (pH7.0), 10% Dextran sulfate, and 50 μg/mL denatured salmon sperm DNA [29] [1 × SSC = 0.15 M NaCl, 0.015 M sodium citrate (pH7.0); 50 × Denhardt’s solution = Ficoll 5 g, polyvinylpyrrolidone 5 g, bovine serum albumin 5 g in 500 mL]. The nitrocellulose membranes were washed with 0.1 × SSC in 0.1% SDS for 20 min at 60 °C for all probes, then exposed to X-ray films (Fuji RX).

### 4.5. Behavior Tests

Open field test: Locomotor activity was measured using an open field test, similar to the previous one [30]. The chamber was made of a square platform with 50 cm × 50 cm × 40 cm (O’Hara and Co. Tokyo, Japan) and illuminated with a light intensity of 100 lux. Mice were placed in the corner of the field and left for 10 min to allow free exploration. During the test, the total distance traveled and the time spent in the central region were recorded and automatically calculated using Image OFCR software (O’Hara and Co., LTD. Tokyo, Japan; see ‘Image analysis for behavioral tests’).

Light/dark transition test: A light/dark transition test was conducted as previously described [31]. The apparatus comprised a cage (21 cm × 42 cm × 25 cm) divided into two sections of equal size by a partition with a door (O’Hara Co.). One chamber was brightly illuminated (light chamber), while the other was not (dark chamber). Mice were placed into the dark chamber and allowed to move freely between the two chambers with the door open for 10 min. The total number of transitions, time spent in each compartment, first latency of movement to the light chamber, and distance traveled were recorded automatically using Image LD software (O’Hara and Co., LTD. Tokyo, Japan; see ‘Image analysis for behavioral tests’).

Elevated plus maze test: Elevated plus maze test was performed as described previously [32]. The elevated plus maze consisted of two open arms (25 cm × 5 cm) and two enclosed arms of the same size, with 15 cm high transparent walls. The arms and central square were made of white plastic sheets, elevated to 60 cm above the floor. To avoid animals falling from the apparatus, 3-mm high Plexiglas sides were used for the open arms. Arms of the same type were arranged on the opposite side. This device was set up under low illumination (center square 100 lux). Each mouse was placed in the central square of the maze (5 cm × 5 cm), facing one of the closed arms, and then behavior was recorded during a 10-min test period. The number of entries and the time spent on open and enclosed arms were recorded. For data analysis, we recorded the percentage of entries onto open arms, the staying time on open arms (seconds), the number of total entries, and the total distance traveled (centimeters). Data acquisition and analysis were performed automatically using Image EP software (O’Hara and Co., LTD. Tokyo, Japan; see ‘Image analysis for behavioral tests’).

Contextual and cued fear conditioning test: Fear conditioning was performed as described previously [33]. Each mouse was placed in a test chamber (33 cm × 25 cm × 28 cm) and allowed to explore freely for 3 min. Then, a 55-dB white noise, which served as the conditioned stimulus (CS), was presented for 20 s, and during the last 2 s of CS presentation, a foot shock (0.2 mA, 2 s), which served as the unconditioned stimulus (US) to mice was given. Two more CS-US pairings were presented with an inter-stimulus interval of 40 s. Twenty-four hours after the conditioning, a contextual test was performed in the same chamber without CS or US stimulus. Forty-eight hours after the conditioning, a cued fear memory was tested in a triangular chamber (33 cm × 33 cm × 32 cm) made of opaque white plastic and allowed to explore freely for 1 min. Subsequently, each mouse was given CS presentation for 3 min. In each session, data acquisition, and control of stimuli (i.e., shocks) were automatically performed, and the percentage of time spent freezing was calculated using Image FZ software (O’Hara and Co., LTD. Tokyo, Japan; see ‘Image analysis for behavioral tests’).

### 4.6. Subcellular Fraction and Western-Blot Analysis

Subcellular fractions were prepared following Carlin’s method [34] with minor modifications. All procedures were performed at 4 °C. Briefly, WT and *Nna1^N^* KO mice were decapitated after cervical dislocation, and the cerebellum and the hippocampus were immediately dissected, removed, and immersed into the homogenization buffer (320 mM sucrose and 5 mM EDTA, containing complete protease inhibitor cocktail tablet (Complete Mini; Roche, Mannheim, Germany) and centrifuged at 1000× *g* for 10 min. The supernatant was centrifuged at 12,000× *g* for 10 min, and the resultant pellet was re-suspended in homogenization buffer at the P2 fraction. The P2 fraction was layered over a 1.2 M/0.8 M sucrose gradient and centrifuged at 90,000× *g* for 2 h. The synaptosome fraction was collected from the interface. The protein concentration was determined using BCA Protein Assay Reagent (Thermo Fisher Scientific Inc. Waltham, MA, USA). An equal volume of SDS sample buffer [125 nm Tris-Cl (pH 6.8), 4% SDS, 20% glycerol, 0.002% BPB, 2% 2-mercaptoethanol] was added to the sample fractions and boiled for 5 min at 95–100 °C.

Protein samples were separated by 7% SDS-PAGE and electro-blotted onto nitrocellulose membranes (GE Healthcare, Piscataway, NJ, USA). Membranes were blocked with 5% skimmed milk in TBS-T [20 mM Tris-Cl (pH 7.6), 137 mM NaCl, 0.1% Tween 20] for 1 h, and were incubated with the following primary antibodies, rabbit polyclonal anti-GluA1 antibody (RRID: AB_2571752; 1:1000, Frontier Institute, Hokkaido, Japan), rabbit polyclonal anti-GluA2 (RRID: AB_2571754; 1:1000, Frontier Institute), rabbit polyclonal anti-GluA3 (RRID:AB_2571598; 1:1000, Frontier Institute), guineapig polyclonal anti-GluA4N (RRID:AB_2571756; 1:1000, Frontier Institute), mouse monoclonal anti-kinesin, heavy chain (RRID:AB_94283; Cat.#MAB1614, clone H2, 1:500, Millipore), mouse monoclonal anti-polyglutamylated tubulin (RRID: AB_477598; clone B3, 1:10000, Sigma-Aldrich, St. Louis, MO, USA), which recognizes the C-terminal region of α-tubulin and β-tubulin, and mouse monoclonal anti-actin antibody (RRID: AB_2223041; clone C4, 1: 2000, Merck Millipore). Each of these primary antibodies was incubated overnight at 4 °C. Tris-buffered saline (10 mM Tris-HCl, pH7.5, 150 mM NaCl) containing 0.1% Tween-20 was used as the dilution medium and washing buffer. Then membranes were incubated for 1 h at around 20 °C with the following peroxidase-conjugated secondary antibodies: anti-rabbit immunoglobulin G (IgG) (RRID: AB_2099233; Cat.#7074, 1:2000, Cell Signaling Technology), anti-mouse IgG (RRID: AB_330924; Cat.#7076, 1: 2000, Cell Signaling Technology) or anti-guinea pig IgG (Cat.#P0141, 1:1000, Dako Cytomation, Glostrup, Denmark). Protein bands were visualized with an enhanced chemiluminescence (ECL) kit (GE Healthcare) using a luminescence image analyzer with an electronically cooled charge-coupled device camera (EZ capture MG; ATTO, Tokyo, Japan). Signal intensities of immunoreacted bands were determined using a CS Analyzer ver.3.0 (ATTO).

### 4.7. Immunohistochemistry

WT and *Nna1^N^* KO mice were deeply anesthetized with sodium pentobarbital by i.p. injection and perfused with 4% paraformaldehyde in 0.1 M phosphate-buffered saline followed by 2 h post-fixation in the same solution. Fresh frozen sagittal sections 20 µm thick were cut with a cryostat. Before immunohistochemical incubation, fresh frozen sections were incubated with 10% normal donkey serum for 20 min, then with a mixture of primary antibodies overnight. We used the following primary antibodies: rabbit polyclonal anti-Car-8 (RRID: AB_2571667; 1 μg/mL, Frontier Institute); Calbindin-D28K antibody (RRID: 1:5000; Swant), rabbit polyclonal anti-GLAST (RRID: AB_2571715; 1 μg/mL, Frontier Institute); rabbit polyclonal anti-GluA1 antibody (RRID: AB_2571752; 1 μg/mL, Frontier Institute); rabbit polyclonal anti-GluA2 (RRID: AB_2571754; 1 μg/mL, Frontier Institute), rabbit polyclonal anti-GluA3 (RRID: AB_2571598; 1 μg/mL, Frontier Institute), guinea pig polyclonal anti-GluA4N (RRID: AB_2571756; 1 μg/mL, Frontier Institute) and mixture of Alexa 488-, Cy3-, or Alexa 647-labeled species-specific secondary antibodies for 2h at room temperature at a dilution of 1:200 (Invitrogen; Jackson Immuno Research). Images were taken with a confocal laser-scanning microscope (FV1000; Olympus, Tokyo, Japan) equipped with HeNe/Ar laser, PlanApo (10×/0.40), PlanApo (20×/0.70), and PlanApoN (60×/1.42, oil immersion) objective lens (Olympus). To avoid overlapping of multiple fluorophores, Alexa 488 and Cy3 fluorescent signals were acquired sequentially using the 488 nm and 543 nm excitation laser lines. All images show single optical sections.

### 4.8. Nissl Staining

Mice were deeply anesthetized with sodium pentobarbital, and then, transcardially perfused with 40 mL of 1% paraformaldehyde in 0.1 mol/L phosphate buffer (pH 7.2). Dissected brains were post-fixed for 2 h in the same fixative at 4 °C. After dehydration, the brain was embedded with paraffin, and sagittal sections 5 µm thick were stained with Cresyl Violet (Nissl) (Invitrogen).

### 4.9. In Situ Hybridization

Mouse cDNA fragments of *Nna1* (nucleotides 272–3754 bp; GenBank accession number, AF219141) were subcloned into the pBluescript II plasmid vector. Digoxigenin (DIG)-labeled cRNA probes were transcribed in vitro [35]. Fragmentation of riboprobes by alkaline digestion was omitted to increase the sensitivity and specificity. For immunohistochemical detection of DIG and fluorescein, sections were blocked with DIG blocking solution [TNT buffer containing 1% blocking reagent (Roche Diagnostics) and 4% normal sheep serum] for 30 min, and 0.5% TSA blocking reagent (PerkinElmer, Norwalk, CT, USA) in TNT buffer for 30 min. Then sections were incubated with peroxidase-conjugated anti-DIG antibody (Roche Diagnostics; 1:1000, 1 h), followed by incubation with the Cy3-TSA plus amplification kit (PerkinElmer). Images were taken with a confocal laser-scanning microscope (FV1000; Olympus, Tokyo, Japan) equipped with HeNe/Ar laser, PlanApo (10×/0.40) and PlanApo (20×/0.70) objective lens (Olympus).

### 4.10. Image Analysis for Behavior Tests

The application software used for the behavioral studies (Image OFCR, LD, EP, and FZ) was based on the public domain NIH Image program (developed by the U.S. National Institutes of Health and available at http://rsb.info.nih.gov/nih-image, accessed on 1 june 2013) and ImageJ program (http://rsb.ingo.nih.gov/ij/, accessed on 1 june 2013), with some modification for each test (available through O’Hara and Co., Japan)

### 4.11. Statistical Analysis

Results were presented as means ± SEM (standard error of the mean). For Western blotting, statistical analyses were performed using Student’s *t*-test. For mouse behavior tests, statistical analyses were performed using Student’s *t*-test. Values in the graphs are expressed as means ± SEM (standard error of the mean). Statistical significance was set at a value of *p* < 0.05. Sample calculation and tests for outliers were not performed.

## Figures and Tables

**Figure 1 ijms-23-12961-f001:**
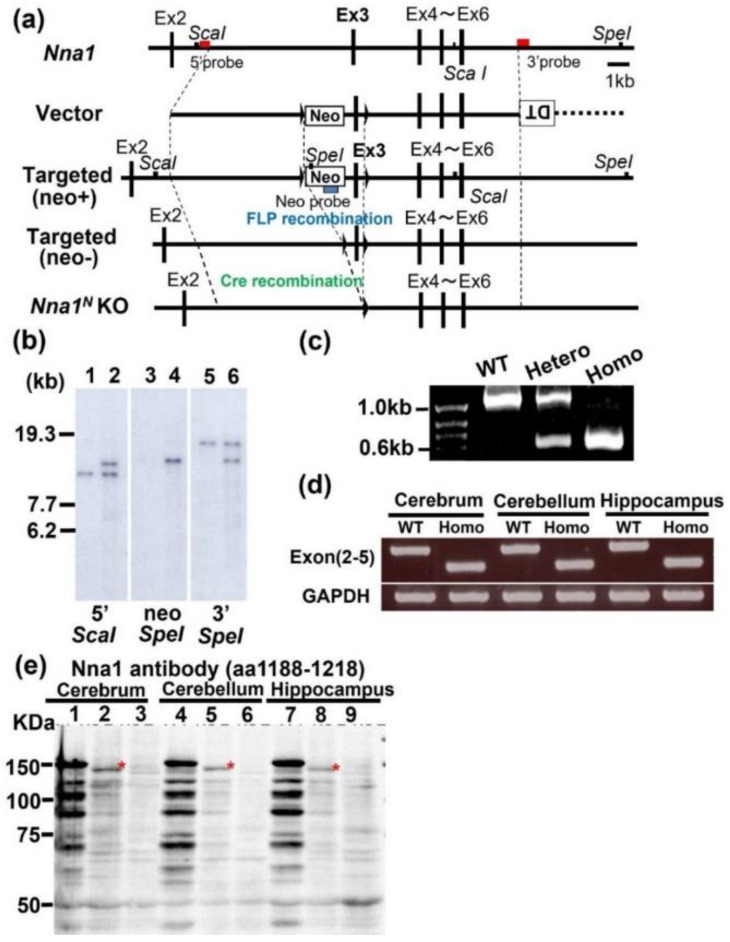
Generation of *Nna1^N^* KO mice. (**a**) Schematic representation of genomic DNA, WT allele, targeting vector, targeted allele, and KO allele after Cre-mediated recombination. Dark boxes represent exon sequences; white open boxes represent neo cassette (Neo) or diphtheria toxin cassette (DT). loxP sites are indicated by triangles. Red bars indicate 5′ or 3′ probe regions used for Southern blot analysis. 5′ outer probe by ScaI, Neo probe and 3′ outer probe by SpeI. (**b**) Southern blot analysis for genomic DNAs from wild-type (WT) and chimeric mice. WT: l, 3, 5, chimera: 2, 4, 6 (n = 3 mice in each genotype). (**c**) Genotypes of WT or *Nna1^N^* KO mice were identified by PCR. (**d**) RT-PCR for detecting exon 3 deletion (n = 3 mice in each genotype). (**e**) Western blot analysis with anti-Nna1 antibody (Frontier Institute). Lanes 1, 4, and 7 indicate WT mice, Lanes 2, 5, and 8 indicate *Nna1^N^* KO mice, and Lanes 3, 6, and 9 indicate *Nna1^C^* KO mice. In the *Nna1^N^* KO brain, lower bands compared to full-length Nna1 protein were detected in *Nna1^N^* KO mice (asterisks) and faint bands in ladder-like patterns (n = 3 mice in each genotype).

**Figure 2 ijms-23-12961-f002:**
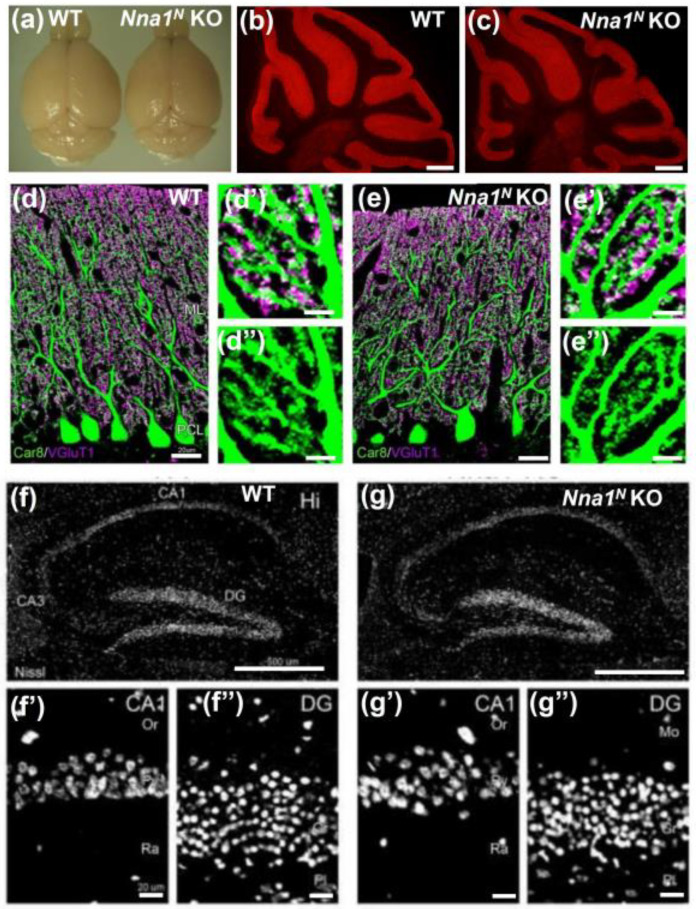
Immunohistochemical observation in the cerebellum and hippocampus of *Nna1^N^* KO mice. (**a**) Macro image of the brains from *Nna1^N^* KO mice and WT mice in the adult age (n = 3 mice in each genotype). No significant atrophy on the parasagittal image of the cerebellum was observed between *Nna1^N^* KO and WT mice. (**b**,**c**) Calbindin-D28K IHC on parasagittal sections (n = 3 mice in each genotype). No significant expression difference was observed in the cerebellum between *Nna1^N^* KO and WT mice (n = 3 mice in each genotype). (**d**,**d′**,**d″**) and (**e**,**e′**,**e″**) Car8 and VGluT1 double IHC also showed no significant difference between KO and WT mice. (**f**,**f′**,**f″**) and (**g**,**g′**,**g″**) Nissl staining on parasagittal sections (n = 3 mice in each genotype). No significant morphological and cell number difference was observed between *Nna1^N^* KO and WT mice. Scale bars: 20 µm in (**d**,**e**,**f′**,**f″**,**g′**,**g″**), Scale bars: 0.5 mm in (**b**,**c**,**f**,**g**), Scale bars: 5 µm in (**d′**,**d″**).

**Figure 3 ijms-23-12961-f003:**
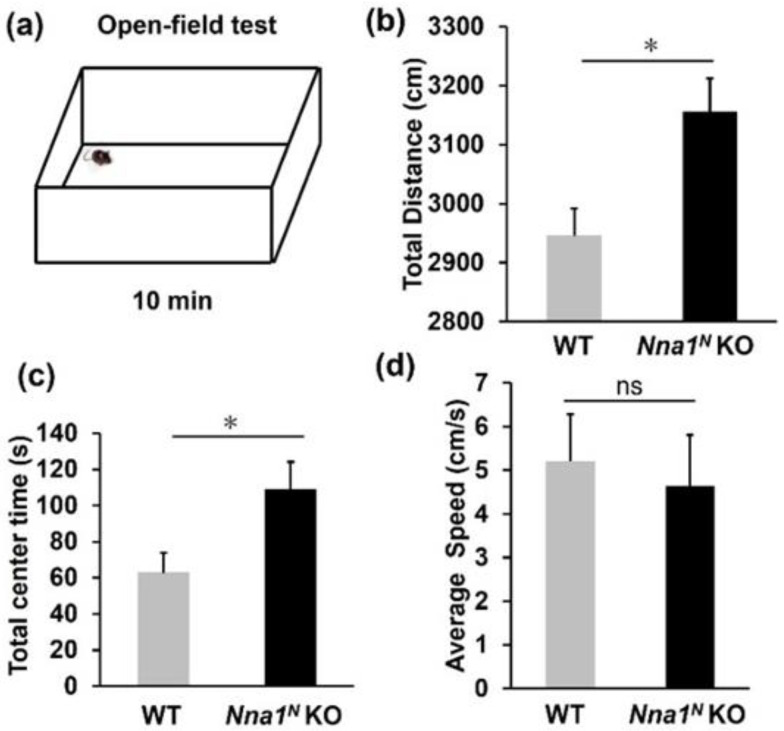
Anxiety-related behavior in *Nna1^N^* KO mice observed by the open-field test. (**a**) Schematic representation of the open-field test. (**b**) *Nna1^N^* KO mice traveled longer than the wild-type (WT) ones in the open-field test. (**c**) *Nna1^N^* KO mice spent a longer time in the central region compared with the WT mice. (**d**) No significant difference was observed in the speed between KO and WT mice. WT, n = 6; *Nna1^N^* KO, n = 10, * *p* < 0.05, Student’s *t*-test. All values presented are means ± SEM. “ns” means not significant.

**Figure 4 ijms-23-12961-f004:**
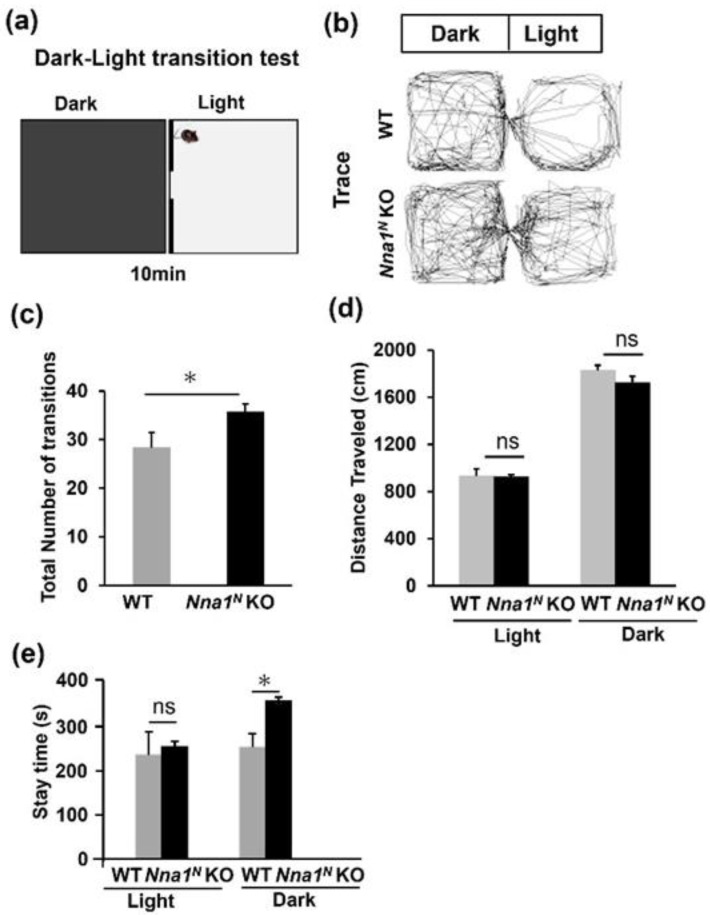
Anxiety-related behavior in *Nna1^N^* KO mice observed by the light-dark transition test. (**a**) Schematic representation of the light-dark transition test. (**b**) Representative traces showing the movement of WT and *Nna1^N^* KO mice separately in the light-dark transition test. (**c**) *Nna1^N^* KO mice showed more transition latency than WT mice. (**d**) No significant difference was observed between *Nna1^N^* KO and WT mice in the travel distance in the boxes. (**e**) *Nna1^N^* KO mice showed more stay time in the dark box than the control mice. WT, n = 6; *Nna1^N^* KO, n = 10, * *p* < 0.05, Student’s *t*-test. All values presented are means ± SEM. “ns” means not significant.

**Figure 5 ijms-23-12961-f005:**
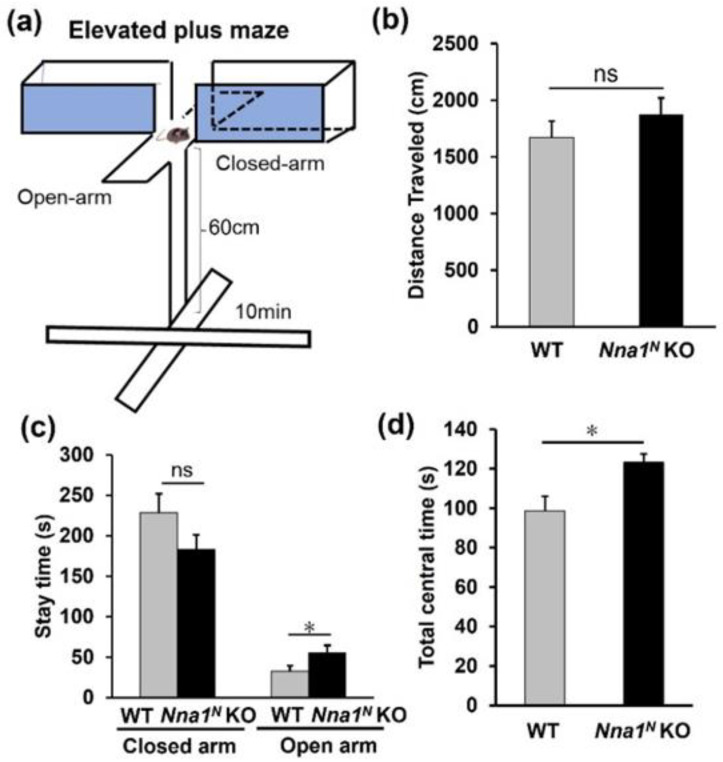
Anxiety-related behavior in *Nna1^N^* KO mice observed by the elevated plus maze test. (**a**) Schematic representation of the elevated plus maze test. (**b**) No significant difference was observed between *Nna1^N^* KO mice and WT ones in the total distance traveled. (**c**) *Nna1^N^* KO mice spent longer time in the open arm compared with WT mice. However, no significant difference in the time spent in the closed arms between KO and WT mice. (**d**) *Nna1^N^* KO mice showed more stay time in the central region than the WT ones. WT, n = 6; *Nna1^N^* KO, n = 10, * *p* < 0.05, Student’s *t*-test. All values presented are means ± SEM. “ns” means not significant.

**Figure 6 ijms-23-12961-f006:**
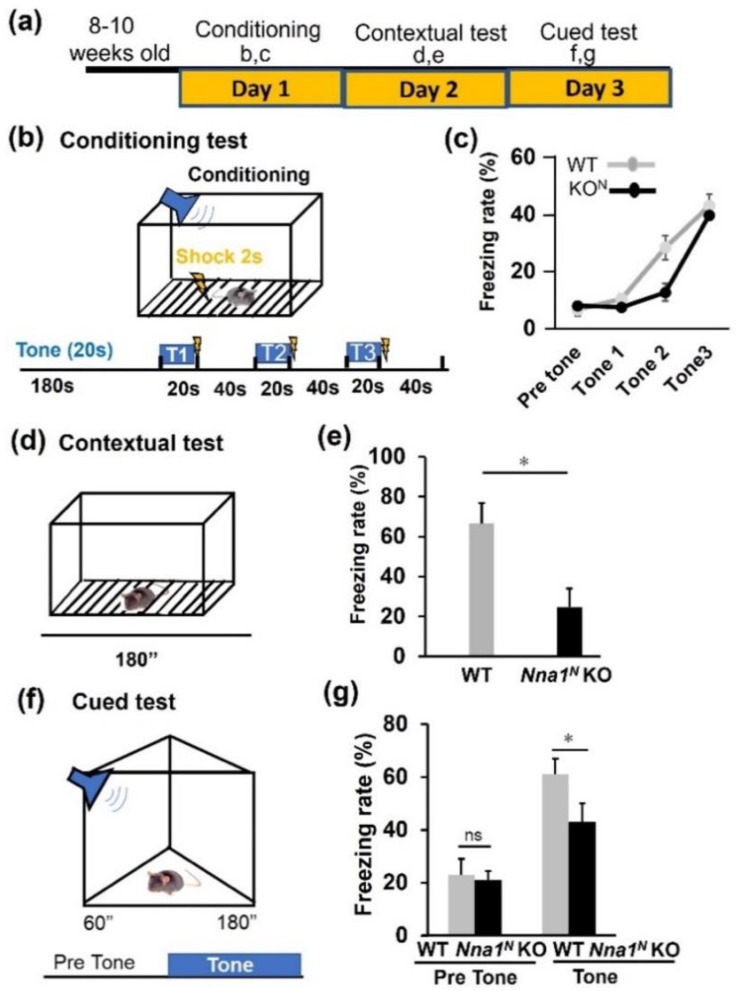
Contextual and cued memory in *Nna1^N^* KO mice in the fear conditioning test. (**a**) Schematic representation of the conditioning test. (**b,c**) There was no significant difference between *Nna1^N^* KO and WT ones in the conditioning tests’ freezing response. (**d,e**) Contextual test: freezing responses on the contextual testing 24 h after conditioning. Note a lower freezing response in *Nna1^N^* KO mice compared with WT mice. *p* < 0.05, Student’s *t*-test. (**f,g**) Cued test: freezing responses on the cued testing 48 h after conditioning. Note the reduced freezing response in *Nna1^N^* KO mice compared to the WT ones. WT, n = 6; *Nna1^N^* KO, n = 10, * *p* < 0.05, Student’s *t*-test. All values presented are means ± SEM. “ns” means not significant.

**Figure 7 ijms-23-12961-f007:**
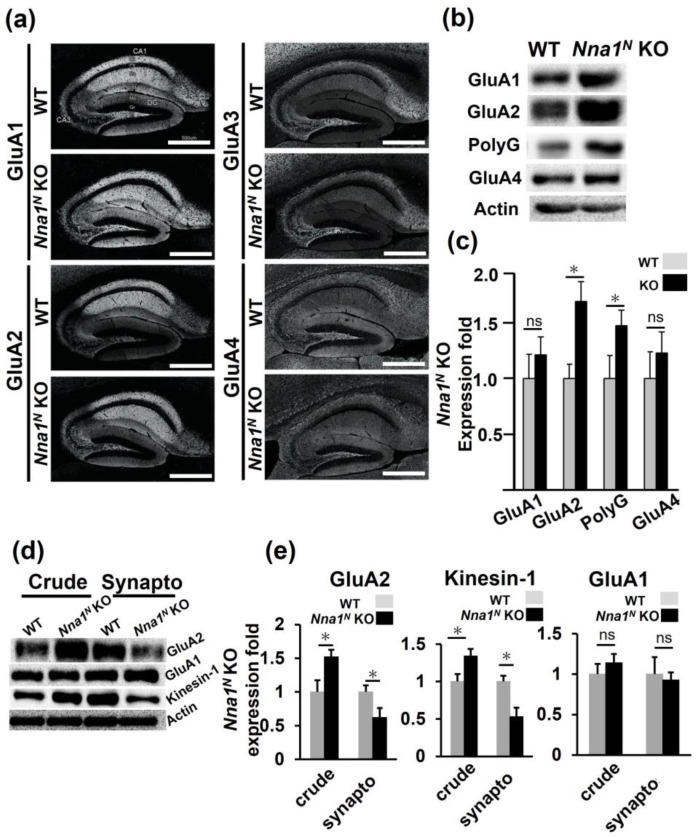
AMPA receptor expression in the hippocampus by immunohistochemistry and Western blotting analysis. (**a**) Protein expression of GluA1–GluA4 in the hippocampus by immunohistochemistry. There was no significant difference between *Nna1^N^* KO mice and WT ones in the expression of GluA1, GluA3, or GluA4 except for GluA2. Scale bars: 20 µM. (**b**,**c**) Crude fraction prepared from the hippocampus was loaded 20 ug each lane to investigate the expression of GluA1, GluA2, GluA4, and Polyglutamylated tubulin (PolyG). Note that increased expression of GluA2 and polyG was observed in the *Nna1^N^* KO mice. All values presented are means ± SEM from 3 experiments. * *p* < 0.05, Student’s *t*-test. (**d**,**e**) kinesin-1 and GluA2 showed a similar expression pattern from the crude fraction (crude) to the synaptosome fraction (synapto). Note that significantly increased expression of GluA2 was observed in the crude fraction (crude) of *Nna1^N^* KO mice. Furthermore, a significant reduction was observed in the synaptosome fraction (synapto) as well kinesin-1. There was no significant modification of GluA1. All values presented are means ± SEM from 3 experiments. * *p* < 0.05, Student’s *t*-test. Β-Actin is the loading control.

**Table 1 ijms-23-12961-t001:** PCR primers used for genotyping and generation of *Nna1* probes.

Name	Forward	Reverse PCR Product
*Nna1^fl^* allele:	5’-CACAGAATCCACACTAATG-3’	5’-GTAAGCACTCCAGACACAC-3’ 1161 bp
*Nna1^wt^* allele:	5’-CACAGAATCCACACTAATG-3’	5’-GTAAGCACTCCAGACACAC-3’ 1050 bp
*Nna1^ΔEx3^* allele:	5’-CATGGACGGGTCCGGGGAGCA-3’	5’-TCAGCCCATCTTCTTCCAGA-3’ 560 bp
*Nna1* (Ex17–23) probe:	5’-AAGCGAGTACGACCTTATC-3’	5’-CCTGTATCCCATGTCTTCCA-3’ 1170 bp
*Nna1* 5’ probe:	5’-CTAGCTCCTCGTTAGAAGTA-3’	5’-GATATAAGTCAACGGTAGA-3’ 373 bp
*Nna1* 3 ‘probe:	5’-TGGTGATGTCCGACCTGTTC-3’	5’-CTTGCAGAAGACCAATCTGCG-3’ 491 bp

## Data Availability

All data generated or analyzed during this study are included in the manuscript and Supporting Files. Raw data have been provided for mean population data shown in figures.

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
