# Peer review of "Nna1*, Essential for Purkinje Cell Survival, Is also Associated with Emotion and Memory"

_ijms, 2022, doi:10.3390/ijms232112961_

Round 1
Reviewer 1 Report
The paper discusses the Nna1/CCP1 (agtbp1) protein was identified as a causative gene for Purkinje cell 34 degeneration (pcd), an autosomal recessive disorder [1]. Defects were detected in model mice. There are 7 figures and all the paper sections are given there are no major spelling/grammar errors thus the paper can be accepted for publication.
More details:
1. Do you consider the topic original or relevant in the field? Does it
address a specific gap in the field?
yes and yes
2. What does it add to the subject area compared with other published
material?
Since the hippocampal function is known to be associated with memory deficits, we 180 examined AMPA-type glutamine receptor subunit expression, which is closely related to 181 hippocampal-dependent behavior
3. What specific improvements should the authors consider regarding the
methodology? What further controls should be considered?
The current methods are ok
4. Are the conclusions consistent with the evidence and arguments
presented and do they address the main question posed?
Conclusions section can be added
5. Are the references appropriate?
The references suffice
Author Response
We greatly appreciate the positive comments of Reviewer1.
Reviewer 2 Report
The investigators examined the neurobiological role of Nna1 in cognitive functions. The data are robust and support the conclusion.
However, I have some queries about statistics:
The authors seemed to use student t test or ANOVA followed by Tukey post hoc test in different types of experiment. As far as I know, t test should be used for comparing 2 groups while ANOVA/post hoc for 3 groups or more. I just wonder why anova/post hoc, instead of t test, was employed for Fig.5b,d, where there are only two groups.
For immunocytochemistry and PCR (Fig.1,2), where results are qualitative, I would like the authors to state how many times the experiments were repeated to yield similar results.
Author Response
We greatly appreciate Reviewer 2 for the positive and constructive comments.
According to the comment, we performed Student’s t-test in Fig5 and described it in the figure legend.
In Fig1 and Fig2, we examined three mice in each genotype for IHC and PCR experiments. Therefore, we added descriptions in Fig1 and Fig2 legends as follows: (n=3 mice in each genotype).